# The Impact of Bariatric Surgery on Quality of Life in Patients with Obesity

**DOI:** 10.3390/jcm12134225

**Published:** 2023-06-23

**Authors:** Radu Petru Soroceanu, Daniel Vasile Timofte, Radu Danila, Sergiu Timofeiov, Roxana Livadariu, Ancuta Andreea Miler, Bogdan Mihnea Ciuntu, Daniela Drugus, Laura Elisabeta Checherita, Ilie Cristian Drochioi, Mihai Liviu Ciofu, Doina Azoicai

**Affiliations:** 1Department of Surgery, “Grigore T. Popa” University of Medicine and Pharmacy, 700115 Iasi, Romania; petru.soroceanu@umfiasi.ro (R.P.S.); radu.danila@umfiasi.ro (R.D.); roxana.livadariu@umfiasi.ro (R.L.); ancuta-andreea_a_miler@d.umfiasi.ro (A.A.M.); bogdan-mihnea.ciuntu@umfiasi.ro (B.M.C.); 2Department of Surgery, “St. Spiridon” County Clinical Emergency Hospital, 700111 Iasi, Romania; 3Department of Preventive Medicine and Interdisciplinarity, “Grigore T. Popa” University of Medicine and Pharmacy, 700115 Iasi, Romania; daniela.drugus@umfiasi.ro (D.D.); doina.azoicai@umfiasi.ro (D.A.); 4Dental Medicine Department, “Grigore T. Popa” University of Medicine and Pharmacy, 700115 Iasi, Romania; checherita.laura@gmail.com; 5Department of Oral and Maxillofacial Surgery, “Grigore T. Popa” University of Medicine and Pharmacy, 700115 Iasi, Romania; ilie-cristian.drochioi@umfiasi.ro (I.C.D.); mihai.ciofu@umfiasi.ro (M.L.C.)

**Keywords:** obesity, bariatric surgery, quality of life, questionnaire, clinical outcomes, LSG, RYGB

## Abstract

Obesity has become a widespread health problem influencing people’s health, general well-being, and healthcare costs. It also represents an important risk factor for multiple comorbidities and malignancies. Objectives: the primary objective of this study was to provide notable insights to healthcare professionals regarding the management of patients with obesity, to highlight the effectiveness of bariatric surgical methods in losing excess weight, and to establish the relationship between weight loss and changes in quality of life (QoL). Material and methods: our study evaluated the QoL of 76 patients following bariatric surgery at different postoperative stages using the 36-Item Short Form Survey (SF-36) and The World Health Organization Quality of Life—BREF (WHOQOL-BREF) questionnaires. Results: regarding the type of bariatric procedure performed, out of the 76 respondents, 39.47% underwent gastric bypass surgery (RYGB), 56.57% underwent sleeve gastrectomy (LSG), and only 3.94% underwent single anastomosis duodeno-ileal switch (SADI-S). Pertaining to the SF-36 questionnaire, the lowest average scores were found in the energy/fatigue subscales and in the limitations due to mental health, which remained consistent across surgery types with a significant decrease in the SADI-S group. Concerning the WHOQOL-BREF questionnaire, the lowest mean scores were found in the environment (15.03 ± 2.37) and social relations (16.08 ± 2.22) subscales, whilst the highest average scores were in physical health (16.30 ± 2.03) and mental health (16.57 ± 2.16). Conclusions: the findings revealed that whilst bariatric surgery significantly improved physical health, it resulted in a decrease in mental health scores. Consequently, the study emphasizes the importance of adopting a holistic approach to managing obesity that considers improving both physical and mental health outcomes.

## 1. Introduction

Obesity has emerged as a pandemic and a serious public health issue due to its high prevalence and detrimental effects on mortality, morbidity, healthcare costs, and QoL [1]. It is now considered a chronic disease [2], and scientific interest in the QoL of individuals with obesity has surged in the past decade. According to estimates, by 2030, over one billion people worldwide will suffer from obesity, affecting one in five women and one in seven men [3]. The most significant population affected by obesity resides in countries with low to middle standards of living, with a predicted twofold increase in obesity rates in low and middle-income countries and a threefold increase in low-income states compared with 2010 reports. This dramatic rise in the prevalence of obesity has prompted global public health research. Current studies demonstrate that QoL decreases inversely proportional to the body mass index (BMI), and individuals with advanced stages of obesity experience more severe declines [4,5]. Regardless of the therapeutic approach, weight loss can improve the QoL in individuals affected by obesity.

Excess body weight is known to substantially diminish the QoL, especially through its impact on overall health and by directly impeding daily activities, resulting in decreased well-being [6]. Furthermore, research on the topic has established that individuals with obesity also experience social stigmatization in addition to physical health consequences [7]. 

Although public health campaigns are crucial in preventing obesity through initiatives such as dietary changes and lifestyle modifications, their effectiveness may be limited for individuals who already have obesity, particularly in severe and complex cases (with a BMI of ≥40 kg/m^2^ or BMI of 35–40 kg/m^2^ accompanied by significant health issues related to excess weight) [8,9]. In such situations, bariatric surgery is considered an optimal treatment option. Research has shown that bariatric surgery not only leads to more substantial weight loss, but also offers better management of type 2 diabetes compared with lifestyle interventions or medication alone [10,11,12]. The two most commonly performed types of bariatric surgery are LSG and RYGB [13].

While recent studies indicate comparable long-term results for both LSG and RYGB, the recurrence of weight gain and comorbidity symptoms in some patients have prompted bariatric surgeons to explore modifications to existing techniques or introduce new ones [14,15]. Among these options, the SADI-S procedure has demonstrated superior effectiveness in achieving long-term weight loss and in remitting comorbidities. However, its technical complexity and potential for adverse events have constrained its widespread adoption [16].

Although multiple studies highlight the positive impact of weight loss on obesity-related comorbidities (both through conservative and surgical means), there are no specific tools, analyses, or questionnaires specifically designed for assessing QoL in patients with obesity [10,17,18,19,20]. 

Therefore, the objectives of our study were to evaluate the overall QoL in patients who underwent different bariatric surgical interventions, to assess specific QoL domains (e.g., physical functioning, mental health, social functioning), to compare the impact of specific bariatric procedures on QoL, and to identify potential predictors or factors associated with significant improvements in QoL following these procedures (such as patient demographics, preoperative conditions, or surgical technique) using the SF-36 and WHOQOL-BREF questionnaires. 

## 2. Materials and Methods

### 2.1. Study Design

The present study was designed as a cross-sectional, non-randomized, and anonymized study based on two commonly used questionnaires used in assessing QoL. The questionnaires were transferred to “Google Forms”. The link was distributed to each individual patient online by email or by telephone. Contact details of our patients were extracted from the database in our bariatric surgery center. 

### 2.2. Inclusion Criteria

The study was implemented in our bariatric surgery center based in the 3rd Surgical Unit at “St. Spiridon” County Clinical Emergency Hospital, Iasi, Romania. Invitations for the survey were sent to 130 patients who underwent a bariatric procedure within the previous 12 months. Incomplete answers or sections of the questionnaires led to the exclusion from the study. Only 76 patients, which included women and men, in various postoperative stages (following LSG, RYGB, or SADI-S) were included in the study. All participants voluntarily completed the questionnaires between 4 January 2023 and 28 February 2023.

### 2.3. The Questionnaires Used

Typically, only a few standardized instruments are used to examine QoL in obese patients. The SF-36 is a self-reported questionnaire that assesses QoL in patients across 8 subscales: physical functionality (PF), limitations in usual role activities (RP), bodily pain (BP), general health (GH), vitality (VT), social functioning (SF), limitations in usual role activities because of emotional problems (RE), and general mental health (MH). The WHOQOL-BREF questionnaire was created by the World Health Organization for the cross-cultural examination of subjective elements related to QoL as an alternative research tool. The 26 items that make up this instrument are divided into 4 categories: physical health (7 items), mental health (6 items), social relationships (3 items), and environmental health (8 items). The physical health domain consists of several items assessing mobility, daily activities, functional capacity, energy, pain, and sleep, while the psychological domain takes into account the perceived self-image, negative thoughts, positive attitudes, self-esteem, mindset, learning ability, memory, concentration, and mental state. The social relationships domain looks at personal relationships, social support, and sex life, while the environmental health field covers financial resources, safety, social and health services, the physical living environment, opportunities for learning and personal development, recreation, noise and pollution, and transportation.

For our cohort, we used the two stated instruments, SF-36 and WHOQOL-BREF, for simultaneous data analysis. The items in the SF-36 questionnaire were translated from English. The patients included in our study speak Romanian as their native language; subsequently, the original questionnaire was compared with its translated version. All translations were performed by two independent certified translators. The questionnaires were finalized and distributed to the patients.

### 2.4. Data Collection and Statistical Methods

Data obtained from the questionnaires were transferred to MS Excel 2010, sourced from Iași, România, “Grigore T. Popa” University of Medicine and Pharmacy. The respondents were classified based on the surgical procedure used (LSG, RYGB, or SADI-S).

The data were then uploaded and processed using the statistical functions in SPSS v. 26.0. All patient data protection provisions were enforced, as the medical team in the teaching hospital was well informed on data and patients’ rights protection. The confidence interval was set at 95% and a *p*-value < 0.05 (two-sided) was considered statistically significant.

In calculating the differences between two or more groups at the 95% significance threshold, depending on the distribution of the value series, descriptive statistics, the Pearson correlation coefficient (r), and the Cronbach α coefficient for internal consistency were used.

## 3. Results

### 3.1. Demographic Characteristics

In our study, the distribution of respondents according to sex indicated an increased frequency of female respondents. Female respondents represented 69.74% and male respondents represented 30.26%.

The study group presented a normal distribution of respondents regarding age, with a maximum frequency of 30.26% corresponding to the 30–39 age group (Table 1).

Regarding the type of bariatric procedure performed, out of the 76 respondents, 39.47% underwent RYGB, 56.57% underwent LSG, and only 3.94% underwent SADI-S. In the group of patients with RYGB (30 respondents), the age ranged from 23 to 69 years, with an average age of 39.47 years (standard deviation = 14.31). Male respondents had a slightly higher average age than female respondents (48.55 vs. 36.47 years) (Table 1). In the group of respondents with LSG (43 respondents), the age ranged from 20 to 69 years, with an average age of 39.97 years (standard deviation = 10.73). Male and female respondents had no significant differences in the mean age. The group of respondents who underwent SADI-S had an average age of 41.33 years (standard deviation = 5.13). The average age was significantly higher among respondents who underwent SADI-S compared with those who underwent RYGB or LSG (41.33 vs. 40.10 vs. 39.97 years).

Out of all the respondents in the study, 39.47% underwent RYGB. Breaking down this percentage by sex, with respect to all respondents, results in 27.63% female and 11.84% male respondents. Among the aforementioned male respondents, 96.66% were under the age of 65. 

Out of all the respondents in the study, 56.57% underwent LSG. Breaking down this percentage by sex, with respect to all respondents, results in 40.79% female and 15.78% male respondents. Among the male respondents, 96.66% were under the age of 65.

Regarding BMI evolution, the obtained results reveal that preoperatively 92.11% of the respondents presented a BMI ≥ 35 kg/m2, and post-operatively only 23.68% presented a BMI ≥ 35 kg/m^2^, finding a correction of the BMI in all groups of patients included in the study (Figure 1). 

### 3.2. Socio-Economic Aspects

Socio-economic aspects can have a major impact on stress and anxiety levels. They can also significantly interfere with lifestyle choices (such as smoking) and have major effects on health, as well as income levels. Regarding income, 50% of patients earn between 2500 and 5000 RON monthly, 27.63% earn more than 5000 RON monthly, and 22.36% earn less than 2500 RON.

### 3.3. Quality of Life Aspects

The SF-36 questionnaire scores (Table 2), along with the Cronbach α coefficient for each subscale, ranged from 0.664 (SF) to 0.704 (PF and MH), indicating good internal consistency. The lowest average scores were found in the vitality subscales (61.13 ± 15.20) and limitations due to mental health (65.78 ± 13.30), which remained consistent across surgery types, with a significant decrease in the SADI-S group (Table 2 and Table 3). Social functioning (89.14 ± 14.76) and self-reported bodily pain (90.49 ± 14.44) had high mean scores (with items related to daily work interference and sickness) compared with others. The general state of health was also assessed.

Table 3 highlights a weak correlation between the mental health subscale (MH) and the SF-36 questionnaire subscales for various surgical procedures. However, a strong correlation exists between the SF (social functionality) subscales and RP (limitations due to physical health), with coefficients of 0.958 and 0.884, respectively. The SADI-S procedure had an average value of 52.00, with a high standard deviation of ±18.33, indicating that the data is scattered from the average. 

The subscales with the highest mean values for a particular surgical intervention are physical pain (92.73 ± 14.23), social functionality (89.53 ± 15.65), and limitations due to physical health (87.20 ± 30.06), which are linked to the LSG procedure. The highest mean scores associated with the RYGB procedure are also included in Table 3.

The WHOQOL-BREF questionnaire had good consistency with Cronbach α coefficients ranging from 0.781 (environment) to 0.845 (social relations). The lowest mean scores were found in the environment (15.03 ± 2.37) and social relations (16.08 ± 2.22) subscales, while the highest average scores were in physical health (16.30 ± 2.03) and mental health (16.57 ± 2.16). Each subscale had similar average and median scores, implying an even distribution (Table 4).

The multivariate analysis of risk factors associated with self-reported QoL on the subscales assessed with the SF-36 questionnaire and the domains of QoL assessed with the WHOQOL-BREF indicates a strong correlation between physical functioning (PF) scores and limitations due to physical health status (RP) (0.000). A significant correlation is also observed between limitations due to physical health status (RP) and social functioning (SF) (0.000) and pain (P) (0.018). Last but not least, physical functionality is closely related to the level of quality of life, with a significant correlation of 0.041. 

Regarding the limitations due to emotional problems (RE), they are strongly associated with the level of vitality (VT) with a significant correlation (0.022). At the same time, the data also shows an association between perceived limitations due to emotional problems with social functioning (SF) (0.005) and the domains of social relations (SR) (0.079) and physical health (PH) (0.002).

Mental health (MH) significantly correlates with general health (GH) (0.12) and social functioning (SF) (0.29). At the same time, mental health is significantly correlated with environmental health, measured as a domain of quality of life (E) (0.016).

The overall trend of mental health (MH) scores compared with the trend of vitality (VT) scores, physical health (PH) scores compared with social functioning (SF) scores, and environmental scores (E) compared with the general trend of scores for social functioning (SF) are illustrated below (Figure 2, Figure 3 and Figure 4).

## 4. Discussion

Recent studies regarding bariatric surgery and its impact on post-surgical patient QoL have primarily focused on biomedical aspects, such as weight loss and improvement of associated pathologies. However, as the amount of such surgical procedures performed worldwide continues to rise, it is crucial to holistically assess the QoL of the patients, including physical and mental performance [21]. Investigations on the topic use the SF-36 instrument to assess these dimensions, but the interpretation of the results is still a matter of debate among experts. In addition, evaluating the QoL for surgical patients may require more specific instruments tailored to certain pathologies, as the SF-36 questionnaire is considered generic and has limitations [22].

A recent study by de Vries (2022) suggests various ways to evaluate post-surgery bariatric patients, both from a clinical and QoL perspective [23]. However, our study also highlighted that scores for physical functionality based on daily activities may not accurately reflect a patient’s QoL. To address these limitations, future research should consider utilizing more specific alternative QoL assessment tools taking into account the specific needs of bariatric patients. Thus, we can better assess the impact of bariatric surgery on the QoL of our patients and identify areas for improvement in post-surgical care.

Bariatric surgery typically leads to pronounced weight loss and improvement of pre-existing diseases, and it can positively affect the QoL of patients, particularly in terms of physical performance, according to a recent study by Albarrán-Sánchez et al. [24]. However, the study also found that mental health scores may decline post-surgery due to factors such as depression, anxiety, eating disorders, unrealistic expectations, low self-esteem, or personality traits. Our study did not focus on these factors, but we did observe low scores in mental health domains similar to the Albarrán-Sánchez study. Upon analyzing the SF-36 scores, we found that the most significant improvements in a patient’s QoL were related to their physical health rather than their mental health.

The Albarrán-Sánchez study also highlighted several factors that may contribute to mental health issues after surgery. First, patients may have a lower sense of overall well-being due to feeling like they are still dealing with a chronic illness, even if they have regained physical function [24]. Our study found that physical function is closely linked to social and emotional well-being. The authors of the study noted that patients may still feel inadequate after surgery, especially because they will continue to have periodic follow-up appointments and may find the lifestyle changes required post-surgery stressful. Additionally, research indicates that accepting morphological changes regarding body image can also cause stress [25]. 

After undergoing bariatric surgery, it is common for patients to experience micronutrient deficiencies. As a result, many patients require regular supplementation of vitamins and minerals, particularly in the case of vitamin B12, which is only obtained from external sources. These deficiencies can be attributed to reduced dietary intake and structural and functional alterations in the gastrointestinal tract, especially in procedures involving malabsorption. However, the occurrence of vitamin and mineral deficiencies following sleeve gastrectomy (LSG) compared with Roux-en-Y gastric bypass (RYGB) has not been documented yet. In a previous study monitoring bariatric patients through the course of 12 months, the authors highlight that the levels of vitamin B12 show a significant decrease, underscoring the importance of long-term supplementation with iron, vitamin B12, and other multivitamins and essential minerals [26]. It should be noted that patients who are not closely monitored may develop anemia due to changes in gastrointestinal absorption [27]. 

Vitamin D deficiency is also very common among bariatric patients. In a clinical study involving patients with obesity, altered basal blood glucose, and hypovitaminosis D, it was observed that correcting vitamin D deficiency through supplementation improves insulin resistance, reduces the risk of developing type 2 diabetes (T2DM), prevents sarcopenia, and regulates adipocyte differentiation [28,29]. Numerous studies have demonstrated the role of vitamin D in modulating the immune system. The low levels of vitamin D in the general population of Europe pose a public health concern, as they have been associated with increased susceptibility to infections and chronic diseases [30]. A recent study found that individuals with low vitamin D levels have an 80% higher likelihood of acquiring a COVID-19 infection compared with a control group with normal levels. 

In addition, regarding mental health and emotional well-being, multiple studies closely link depression and obesity. Both pathologies are considered risk factors for one another and tend to associate within individuals. They also seem to have shared biological mechanisms [31]. Recent scientific research, as demonstrated by studies conducted by Robinson et al. and Sullivan et al., provides evidence supporting the influence of genetic factors on both obesity and depression [32,33]. These studies indicate that there is an estimated 40% heritability for major depressive disorder (MDD) and BMI. Furthermore, through genome-wide association analyses, Pigeyre et al. identified over 200 genomic regions associated with BMI, obesity status, and fat distribution. It seems that the genes located in proximity to BMI-associated loci show significant expression in the hypothalamus and pituitary gland [34]. These brain regions are responsible for regulating both mood and energy homeostasis. Additionally, more than 50 genetic loci associated with depression phenotypes, specifically related to MDD genetics, have been identified [35]. Wheeler and Pierce note that the genes exhibiting the strongest signals were previously linked to severe early-onset obesity and are situated in close proximity or even have overlapping positions [36]. 

The results of a study published in the Journal of the American Medical Association (JAMA) indicate that a substantial number of young individuals with severe obesity who underwent bariatric surgery reported a reduction of over 50% in their excess body weight, resulting in significant improvements in health and QoL compared with young individuals who were engaged in an intensive lifestyle modification program [37]. Obesity is a major but manageable risk factor for many diseases. Additionally, current research consistently demonstrates that obesity has a negative impact on QoL, and the severity of these effects escalates proportionally with the level of obesity, which aligns with the findings of our study [38].

Patients with severe obesity have been reported to have a lower QoL, particularly in regard to mental health, compared with normal-weight patients [39]. However, physical functionality has the greatest impact on perceived QoL [40]. Bariatric surgery has been found to improve various aspects of QoL, including patient satisfaction, self-esteem, body perception, and social interaction, in addition to weight loss [39]. Our study found a close relationship between physical and mental health. The latter (MH) significantly correlates with general health (GH). Despite the significant weight loss in the first year following bariatric surgery, few prospective studies have evaluated the long-term effect of depression, anxiety, and QoL, as well as their effects on weight loss recovery [24].

A recent meta-analysis indicates that RYGB and LSG may improve the post-operative QoL of the patients. The study assesses patients at various time points, from one up to five years after surgery [40]. Our study found that patients who underwent RYGB or LSG reported a higher self-perceived QoL when compared with those who underwent SADI-S, as measured using the SF-36 questionnaire. The meta-analysis also suggests that more complex or invasive procedures may result in lower QoL in the short and medium term, but improvement is seen after nine years.

Our study also found that evaluating the QoL of bariatric surgery patients is challenging, despite the weight loss and improvement of coexisting illnesses. Research in the literature emphasizes the importance of including patient-reported QoL assessment in the therapeutic approach, as it provides valuable data for evaluating the success of surgery beyond clinical evaluation. Our data highlighted that patients report improved physical functionality, which significantly correlates with mental health. However, for accurate measurements, longitudinal studies are necessary to measure the perception of QoL in physical health, mental health, and social relationships.

Potential shortcomings of our study include the risk of selection bias, as the respondents were volunteers. This could limit the applicability of the findings to the entire population of bariatric patients. Without a control group, it could be challenging to determine whether the observed improvements in quality of life can be solely attributed to bariatric surgery. As in all other studies relying on self-reported data from questionnaires, some of the respondents might be subject to recall bias or subjective interpretation. The study has a relatively short follow-up duration. This could restrict the ability to assess the long-term effects and sustainability of improved quality-of-life outcomes.

The study utilized the SF-36 and WHOQOL-BREF questionnaires, which are widely recognized and validated tools for measuring quality of life. This adds credibility to the study’s findings and provides a structured and objective approach to evaluating the impact of bariatric surgery on quality of life.

With 76 respondents, the study had a reasonable sample size, which enhances the statistical power and reliability of the results. Assessing patients at different timepoints after the surgical procedure allows for a comprehensive understanding of the overall impact on the quality of life.

## 5. Conclusions

The current study provides insight into the QoL of patients who underwent different types of bariatric procedures. The findings indicate that following surgery, there was an improvement in the physical and social functioning of patients. The study also points out areas where patients may still encounter difficulties, such as issues related to vitality and limitations associated with mental health. The study can provide valuable insights for medical professionals to develop targeted interventions aimed at improving patient outcomes following bariatric surgery. Additionally, it contributes to the growing body of evidence concerning the impact of bariatric surgery on QoL, thereby guiding future research in this field.

## Figures and Tables

**Figure 1 jcm-12-04225-f001:**
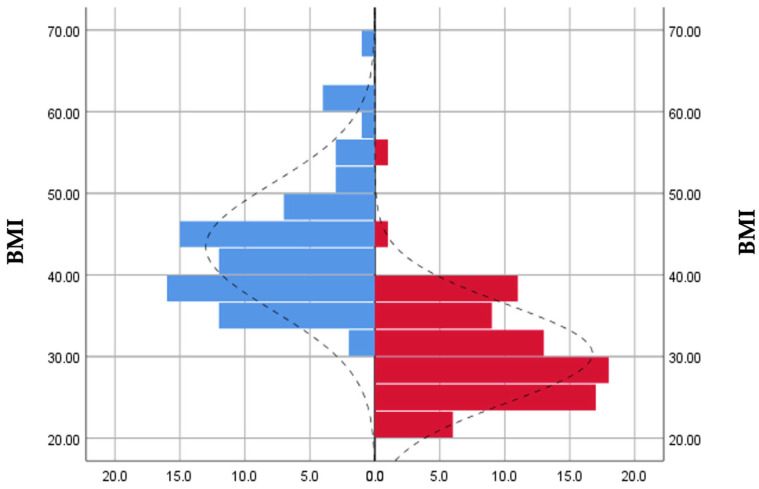
Preoperative (blue) and postoperative (red) BMI distribution.

**Figure 2 jcm-12-04225-f002:**
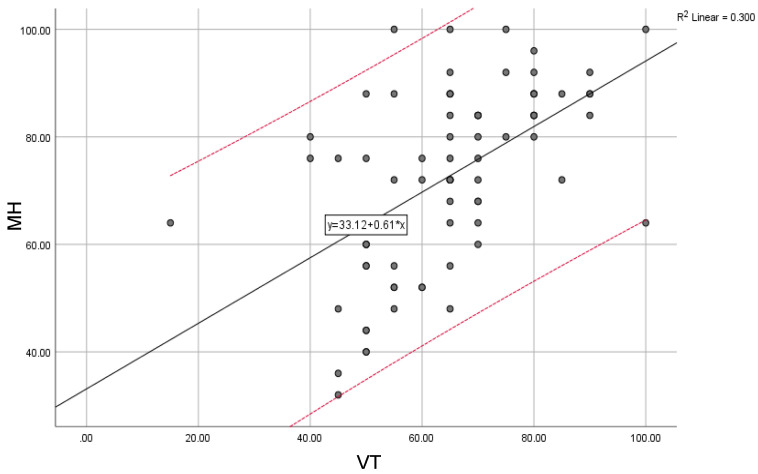
Mental health (MH) compared with the general trend for vitality (VT) scores; “y” indicates the overall trend of the scores.

**Figure 3 jcm-12-04225-f003:**
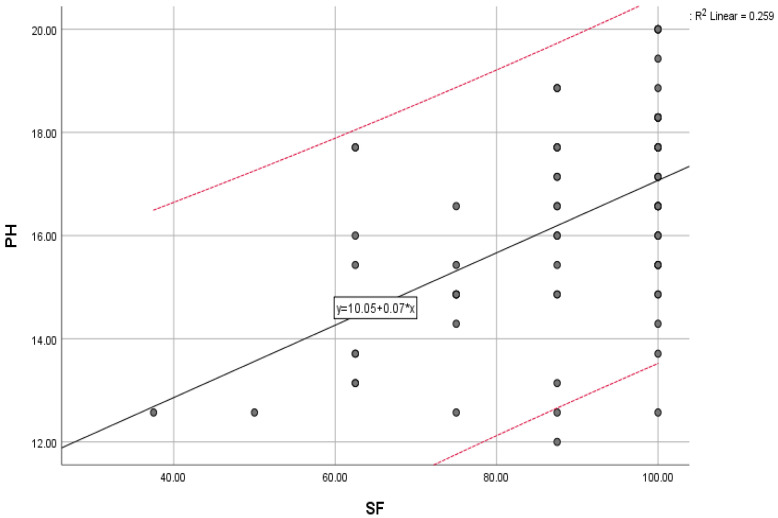
Physical health scores (PH) compared with the general trend for social functioning (SF) scores; “y” indicates the overall trend of the scores.

**Figure 4 jcm-12-04225-f004:**
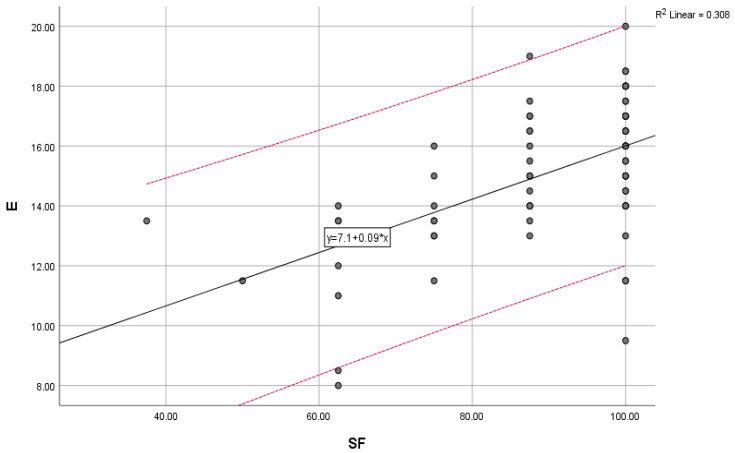
Environmental scores (E) compared with the general trend for social functioning (SF) scores; “y” indicates the overall trend of the scores.

**Table 1 jcm-12-04225-t001:** Descriptive characteristics regarding the age of respondents according to the type of surgical procedure and sex.

	Sex	n	%	Mean Age	Std. Dev.	Min. Age	Max. Age	*p*
**RYGB**	Women	21	27.63	36.47	13.01	23	65	0.269
	Men	9	11.84	48.55	14.31	26	69	
	Total	30	39.47	40.10	14.31	23	69	
**LSG**	Women	31	40.79	39.35	10.22	20	69	
	Men	12	15.78	41.58	12.28	27	69	0.441
	Total	43	56.57	39.97	10.73	20	69	
**SADI-S**	Women	1	1.31	-	-	-	-	-
	Men	2	2.63	42.00	7.07	37	47	
	Total	3	3.94	41.33	5.13	36	47	

**Table 2 jcm-12-04225-t002:** Statistical indicators regarding the SF-36 questionnaire score for each subscale.

Subscale	Avg.	SD	Mean	Min.	Max.	Q25	Q75	Cronbach α
**PF**	84.01	25.82	95.00	0.00	100.00	82.50	100.00	0.704
**RP**	85.85	30.09	100.00	0.00	100.00	100.00	100.00	0.667
**RE**	65.78	13.30	66.66	33.33	100.00	66.66	66.66	0.672
**VT**	61.13	15.20	65.00	15.00	100.00	52.50	75.00	0.674
**MH**	68.65	16.92	76.00	32.00	100.00	60.00	88.00	0.704
**SF**	89.14	14.76	100.00	37.50	100.00	87.50	100.00	0.664
**BP**	90.49	14.44	100.00	45.00	100.00	78.75	100.00	0.688
**GH**	76.52	16.22	80.00	31.25	100.00	65.00	90.00	0.685

Physical functionality (PF), limitations in usual role activities (RP), limitations in usual role activities because of emotional problems (RE), vitality (VT), general mental health (MH), social functioning (SF), bodily pain (BP) and general health (GH)

**Table 3 jcm-12-04225-t003:** Mean values of the SF-36 questionnaire score according to each subscale and type of surgical intervention.

	RYGB	LSG	SADI-S	*p*
**PF**	82.33 ± 26.21	84.41 ± 26.48	95.00 ± 8.66	0.717
**RP**	82.50 ± 31.58	87.20 ± 30.06	100.00 ± 0.00	0.577
**RE**	66.66 ± 15.16	65.11 ± 12.50	66.67 ± 0.00	0.884
**VT**	63.50 ± 15.81	65.81 ± 15.11	58.33 ± 11.54	0.631
**MH**	71.33 ± 14.78	74.79 ± 17.55	52.00 ± 18.33	0.068
**SF**	88.75 ± 14.06	89.53 ± 15.65	87.50 ± 12.50	0.958
**BP**	87.41 ± 14.30	92.73 ± 14.23	89.16 ± 18.76	0.302
**GH**	70.12 ± 17.07	75.94 ± 15.96	80.00 ± 13.22	0.652

Physical functionality (PF), limitations in usual role activities (RP), limitations in usual role activities because of emotional problems (RE), vitality (VT), general mental health (MH), social functioning (SF), bodily pain (BP) and general health (GH)

**Table 4 jcm-12-04225-t004:** Descriptive statistics of the score obtained from the WHOQOL-BREF questionnaire based on each domain.

Domains	Avg.	SD	Mean	Min.	Max.	Q25	Q75	Cronbach α
**Physical health**	16.30	2.03	16.57	12.00	20.00	14.85	17.71	0.835
**Mental health**	16.57	2.16	16.66	10.00	20.00	15.33	18.33	0.782
**Social** **relationships**	16.08	2.22	16.00	12.00	20.00	14.66	17.33	0.845
**Environment**	15.03	2.37	15.00	8.00	20.00	13.50	17.00	0.781

## Data Availability

The data presented in this study are available on request from the corresponding author.

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
