# Peer review of "The Impact of Bariatric Surgery on Quality of Life in Patients with Obesity"

_jcm, 2023, doi:10.3390/jcm12134225_

Round 1

Reviewer 1 Report

The results reported in the present manuscript are valuable, however several major issues have to be adressed before considering for publication.

1. The manuscript has to be prepared in accordance to STROBE checklist for cohort studies.

2. In main text, every abbreviation has to be explained the first time that is used.

3. Introduction section should be shorter with less details regarding obesity in general.

4. The last paragraph of introduction section should report the aim of the study.

5. The whole materials and methods section should be reformed in accordance to STROBE.

6. In results section, Table 1 and 2 as well as figure 1 are unnecessary and should be removed.

7. Please add footnotes in Tables 4 and 5.

8. Discussion section is not well-written and the connection of stydy's results with the literature is not well-documented.

9. Useful literature data that can be used for discussion section are reported in a recently published chapter (DOI:10.1007/978-3-031-27597-5_5)

10. Conclusion section should be shorter.

An English language review is recommended.

Author Response

Dear reviewer, please find the answers in the atachement

Reviewer 2 Report

An interesting study by Soroceanu et al. that investigates the impact of bariatric surgery on patients' QoL. Despite its clinical significance, this topic is not systematically addressed in bariatric research studies, thus inhibiting the extrapolation of hard evidence. This original paper depicts the effect of bariatric surgery on perioperative QoL, based on universally validated tools (SF 36 and WHOQOL-BREF).

Overall the manuscript is well structured with up-to-date references. Moreover the tables are of high quality and quite informative. A few comments, though, prior to publication:

-A linguistic review by a native English speaker should be considered.

-Please reduce the length of the abstract and introduction section.

-The study aims should be clearly reported.

-A study design section should be provided in the methods section. The authors should provide information regarding the study protocol and the ethics comittee clearance. Moreover data regarding the patient recruitment method should be reported.

-Furthermore it should be clearly stated whether the study was a prospective or a retrospective cohort, alongside with the data collection methodology.

- A statistical analysis methodology should be adequately described

-Please provide the results in a more concise way

-Figure 3,  4, 5 were not provided

-Please provide a summary table with the QoL measurements over all time periods.

Minor linguistic and phraseological revisions are required to cope with the journal publication standards.

Author Response

Dear reviewer, please find the response in atache file. Thank you!

Round 2

Reviewer 1 Report

Authors made an effort to improve their manuscript.

Minor editing.

Reviewer 2 Report

The authors successfully addressed all comments raised by the authors